# Bacterial and Genetic Features of Raw Retail Pork Meat: Integrative Analysis of Antibiotic Susceptibility, Whole-Genome Sequencing, and Metagenomics

**DOI:** 10.3390/antibiotics13080700

**Published:** 2024-07-26

**Authors:** Michelle Lowe, Wilhelmina Strasheim, Wai Yin Chan, Olga Perovic

**Affiliations:** 1Centre for Healthcare Association Infections, Antimicrobial Resistance and Mycoses, National Institute for Communicable Diseases, a Division of the National Health Laboratory Service, Johannesburg 2192, South Africa; michelle.lowe@wits.ac.za (M.L.); wilhelminas@nicd.ac.za (W.S.); 2Department of Clinical Microbiology and Infectious Diseases, School of Pathology, University of Witwatersrand, Johannesburg 2193, South Africa; 3Sequencing Core Facility, National Institute for Communicable Diseases, a Division of the National Health Laboratory Service, Johannesburg 2192, South Africa; 4Department of Biochemistry, Genetics and Microbiology, Forestry and Agricultural Biotechnology Institute, University of Pretoria, Pretoria 0002, South Africa

**Keywords:** whole-genome sequencing, metagenomics, antibiotic residue testing, retail pork meat, South Africa

## Abstract

The global antibiotic resistance crisis, driven by overuse and misuse of antibiotics, is multifaceted. This study aimed to assess the microbiological and genetic characteristics of raw retail pork meat through various methods, including the isolation, antibiotic susceptibility testing (AST), whole-genome sequencing (WGS) of selected indicator bacteria, antibiotic residue testing, and metagenomic sequencing. Samples were purchased from 10 pre-selected retail stores in Gauteng, South Africa. The samples were aseptically separated, with portions sent to an external laboratory for isolating indicator bacteria and testing for antibiotic residues. Identification of the isolated bacteria was reconfirmed using matrix-assisted laser desorption/ionization time-of-flight mass spectrometry (MALDI-TOF MS). AST was performed using the Microscan Walkaway system (Beckman Coulter, Brea, CA, USA). WGS and metagenomic sequencing were performed using the Illumina NextSeq 550 instrument (San Diego, CA, USA). The isolated *E. coli* and *E. faecalis* exhibited minimal phenotypic resistance, with WGS revealing the presence of tetracycline resistance genes. Both the isolated bacteria and meat samples harboured tetracycline resistance genes and the antibiotic residue concentrations were within acceptable limits for human consumption. In the metagenomic context, most identified bacteria were of food/meat spoilage and environmental origin. The resistome analysis primarily indicated beta-lactam, tetracycline and multidrug resistance genes. Further research is needed to understand the broader implications of these findings on environmental health and antibiotic resistance.

## 1. Introduction

Since 2005, pork consumption in South Africa has seen a significant increase [1,2]. The country has a well-established pork production industry, with the largest contributions coming from the Limpopo and North West provinces, accounting for 24% and 20% of the total production, respectively [1]. In South Africa, meat is sold through formal and informal channels, including butcheries, supermarkets, farms, and open markets [2]. The formal sector adheres to safety checks before selling meat to consumers; informal outlets often skip these checks, posing risks of microbial contamination, antibiotic resistance, and poor meat quality [2].

Antibiotic resistance is multifaceted and has been linked to overuse and misuse of antibiotics in humans and animals, improper prescribing practices, and the widespread use of antibiotics in livestock production [3,4]. Antibiotic resistance is a critical concern in pork production and other food production systems, as it can negatively impact human health [5]. Antibiotic-resistant bacteria and antibiotic resistance genes (ARGs) can contaminate food at any stage, from field to retail [2,5]. Previous studies have found that food products not only serve as a reservoir for antibiotic-resistant bacteria and ARGs but also act as a mediator, facilitating the transfer of antibiotic-resistant bacteria and ARGs between the environment and humans through the consumption of contaminated foods [4,6,7,8]. ARGs can also be shared among different bacterial species through horizontal gene transfer [9]. Among these species are commensal flora and pathogenic foodborne pathogens, including *Campylobacter* spp., *Enterococcus* spp., *Escherichia coli*, and non-typhoidal *Salmonella* spp. (NTS), which can cause diseases in both humans and animals [2,9].

In animals, antibiotics such as sulphonamides, tetracyclines, and fluoroquinolones are widely used for therapeutic, metaphylactic, and prophylactic purposes or as growth promotors in animal feed [8,10,11]. Understanding the diversity and abundance of ARGs, virulence factor (VF) genes, and antibiotic residues in food, especially retail pork meat, is important for controlling antibiotic resistance [4,10]. This involves implementing effective antibiotic stewardship programs, regulating the use of antibiotics in food production, improving hygiene and sanitation practices, and promoting responsible antibiotic usage in both human and veterinary medicine [8]. To address this issue, the Food and Agriculture Organization and the European Union have established tolerance levels, known as maximum residue limits (MRLs), for antibiotic residues in animal-derived food products [12]. In South Africa, these MRLs are regulated by the Foodstuffs, Cosmetics and Disinfectants Act (Act No. 54 of 1972) and the Meat Safety Act (Act No. 40 of 2000) [13,14].

The aim of this study was (1) to isolate and characterise four common indicator bacteria, *Campylobacter* spp., *Enterococcus* spp., *E. coli*, and NTS, that overlap between humans and animals; (2) to test for antibiotic residues; and (3) to assess the microbial community as well as the resistome present in purchased raw retail pork meat.

## 2. Results

### 2.1. Isolation of Indicator Bacteria

All 10 raw meat samples were tested for the selected four indicator bacteria (Table 1). *E. coli* was detected and isolated from one butchery raw meat sample (PC9-B4). *E. faecalis* were detected and isolated from two supermarket raw meat samples (PC3-S3 and PC4-S4) and one butchery raw meat sample (PC10-B5). *Campylobacter* spp. and *Salmonella* spp. were not detected in any of the collected raw meat samples (Appendix A).

### 2.2. Antibiotic Susceptibility Testing (AST) and Whole-Genome Sequencing (WGS) of Isolated Indicator Bacteria

A single *E. coli* isolate was obtained from sample PC9-B4, while one *E. faecalis* isolate was obtained from each of the samples PC3-S3, PC4-S4, and PC10-B5. The IDs of all the isolates were reconfirmed and AST was performed. All three *E. faecalis* isolates had the same AST profile (Table 2). There are no AST guidelines for isolates obtained from raw meat samples; thus, the AST results were interpreted using the 2023 EUCAST guidelines for human isolates [15].

The isolated *E. coli* (PC9-B4) and *E. faecalis* (PC3-S3, PC4-S4, and PC10-B5) isolates were subjected to WGS (Table 3). The *E. coli* isolate exhibited ARGs conferring resistance to aminoglycoside (*aad*A1), fluoroquinolone (*gyr*A), tetracycline (*tet*B), trimethoprim (*dfr*A1), and sulphonamide (*sul*2). All three *E. faecalis* isolates exhibited ARGs conferring resistance to tetracycline (*tet*M) and lincosamides (*isa*A).

A total of eight VF genes were identified in *E. coli*, and nine VF genes were identified in *E. faecalis.* VF genes associated with *E. coli* included colicins (*cba*, *cea*, *cia*, and *cma*), as well as the toxin gene (*ast*A). VF genes associated with *E. faecalis* isolates included adhesins (*ace* and *efaAfs*), toxins (*cyl*-A, -L, and -M), and genes associated with biofilm formation (*ebp*-A and *-B*) and pheromone production (*cad*, *cam*E, *cCF10*, and *cOB1*).

The *E. coli* isolate harboured the IncB/O/K/Z and IncFII (pCoo) plasmids, while the *E. faecalis* isolates harboured the repUS43, repUS11, and rep9a plasmids.

### 2.3. Antibiotic Residue Testing

The average antibiotic residue concentration detected in all the raw meat samples was less than 50 µg/kg for the majority of the tested antibiotics (Table 4). Sample PC5-S5 had an antibiotic residue concentration of 71.5 µg/kg for chlortetracycline. All tested residue levels were below the acceptable limits set by Codex/SA MRL [12].

### 2.4. Metagenomics

#### 2.4.1. Read Statistics

Approximately 7 million host reads (ranging from 5,797,550–8,900,728 sequencing reads) were removed from each sample prior to the metagenomic analysis (Table 5). The remaining reads were subject to further bacterial community profiling and ARG prediction analyses.

#### 2.4.2. Estimated Relative Abundance

The distribution of the bacterial community structures in the raw meat samples reveals functional diversity among bacteria (Figure 1). The relative abundance was similar between raw meat samples from supermarkets and butcheries, but PC9 was different.

Food and meat spoilage organisms such as *Pseudomonas* (90%; 9/10), *Acinetobacter* (80%; 8/10), *Brochothrix* (80%; 8/10), *Psychrobacter* (70%; 7/10), *Photobacterium* (60; 6/10), and *Clostridium* (50%; 5/10) were detected in the majority of raw meat samples [16,17,18,19,20].

#### 2.4.3. Resistome Prediction

Using the assembled contigs generated by MEGAHIT from the 10 raw meat samples, the PathoFact pipeline predicted a total of 61,665 ORFs. Among these, 89 ORFs were associated with ARGs, 138 with VF secretion, and 40 with toxin secretion (Appendix A).

##### Antibiotic Resistance Gene Prediction

A total of 89 ORFs were predicted to be ARGs (Table 5; Appendix A). The raw meat samples from supermarkets were dominated by multidrug (26.8%; 18/67), beta-lactam (20.9%; 14/67), and tetracycline (14.9%; 10/67) resistance genes. The raw meat samples from butcheries were dominated by tetracycline (36.4%; 8/22) and multidrug (13.6%; 3/22) (Figure 2 and Figure 3A,C) resistance genes.

The majority of the ARGs were involved with resistance mechanisms such as antibiotic efflux (48.3%; 43/89), antibiotic inactivation (23.6%; 21/89), and antibiotic target alteration (12.36%; 11/89) (Appendix A). No ARGs were predicted in sample PC6 (butchery—B1) (Figure 3B,D; Appendix A).

##### Virulence Factor and Toxin Gene Prediction

A total of 138 ORFs were predicted to be secreted VF genes. These were categorised as follows: adherence (36.9%; 51/138), antimicrobial activity/competitive advantage (2.9%; 4/138), biofilm (5.1%; 7/138), effector delivery system (13.8%; 19/138), exoenzyme (0.7%; 1/138), exotoxin (0.7%; 1/138), immune modulation (8%; 11/138), invasion (5.1%; 7/138), motility (9.4%; 13/138), nutritional/metabolic factor (10.9%; 15/138), post-translational modification (1.4%; 2/138), regulation (1.4%; 2/138), and stress survival (3%; 4/138) (Table 5; Figure 4A; Appendix A).

VF genes from all categories were present in the raw meat samples collected from supermarkets (Appendix A). No exotoxin, invasion, post-translational modification, or stress survival VF genes were detected in the raw meat samples collected from butcheries. Fifteen types of toxin genes were detected in the raw meat samples collected from supermarkets and butcheries (Figure 4B).

## 3. Materials and Methods

### 3.1. Ethical Clearance and Study Definitions

The study was conducted in accordance with the Declaration of Helsinki and approved by the Human Research Ethics Committee of the University of the Witwatersrand (M190244; 10 May 2019).

Supermarkets were defined as stores that sold raw meat and various other grocery items, while butcheries were defined as stores selling primarily raw meat commodities. Raw meat samples were defined as pork chops (alternative names included loin, rib, sirloin, top loin, and blade chops). Indicator bacteria in this study were defined as *Campylobacter* spp., *Enterococcus* spp., *E. coli*, and NTS spp.

### 3.2. Study Setting and Sampling

Raw meat samples (i.e., pork chops) were purchased on the 4 January 2022, from 10 pre-selected supermarkets and butcheries in Johannesburg and Pretoria, Gauteng. Convenience sampling was used based on the location of the laboratory, Centre for Healthcare-Associated Infections, Antimicrobial Resistance and Mycoses (CHARM), National Institute for Communicable Diseases (NICD). The stores selected for sample collection included both supermarkets (*n* = 5) and butcheries (*n* = 5). Pork chops were chosen as it is the most popular cut among consumers [21]. Raw meat samples (containing at least two pork chops in the same container/packet) were randomly selected from the store shelves. All the raw meat samples were collected within the recommended dates for human consumption. Store demographics were captured by the study investigators on the day of sample collection (Appendix A). A unique number was assigned to each raw meat sample as well as the sampled stores to maintain anonymity and to ensure that the results cannot be linked back to a specific store.

The purchased raw meat samples were transported on ice to CHARM, NICD. The outsides of all the meat containers/packets were wiped with 70% ethanol before segregation and processing to avoid cross-contamination. The raw meat samples were separated aseptically. One raw meat sample from each store was sent on ice within 24 h after collection to a subcontracted laboratory in Gauteng, South Africa, for (1) the isolation of four selected indicator bacteria, and (2) antibiotic residue testing using liquid chromatography–tandem mass spectrometry (LC-MS/MS). The remaining raw meat samples underwent metagenomics sequencing at the Sequencing Core Facility (SCF), NICD. All the raw meat samples were processed within 24 h to 48 h after sample collection.

### 3.3. AST and WGS of Isolated Indicator Bacteria from Raw Meat Samples

The bacteria isolated from the raw meat samples were transported at room temperature (20 °C to 25 °C) from the subcontracted laboratory to CHARM, NICD within 24 h after isolation. Organism identification was reconfirmed at CHARM, NICD, with matrix-assisted laser desorption/ionization time-of-flight mass spectrometry (MALDI-TOF MS) (Microflex, Bruker Daltonics, Bremen, Germany). AST was performed using the Microscan Walkaway System with the Gram-negative NM44 (Beckman Coulter, Brea, CA, USA) and Gram-positive PM33 (Beckman Coulter, USA) cards. The AST results were interpreted using the 2023 European Committee on antimicrobial susceptibility testing (EUCAST) guidelines [15].

The genomic DNA (gDNA) of four isolated organisms from the raw meat samples was extracted with the QIAamp mini kit (Qiagen, Hilden, Germany) with the inclusion of lysozyme (10 mg/mL; Sigma-Aldrich, St. Louis, MO, USA) to ensure sufficient lysis. The quantity and quality of the extracted gDNA were determined on Qubit 4.0 (Invitrogen, Pittsburgh, PA, USA) with the high sensitivity assay kit (Invitrogen, USA). The tagging and fragmentation of the gDNA were performed using the Nextera DNA Flex Library preparation kit. Multiplexed paired-end libraries were prepared using the Nextera DNA preparation kit, followed by sequencing (2 × 150 bp) on a NextSeq 550 instrument (Illumina Inc., San Diego, CA, USA) with 100× coverage at the SCF, NICD. Raw paired-end reads were analysed using the Jekesa pipeline (v1.0; https://github.com/stanikae/jekesa (accessed on 3 April 2023) [22]. Briefly, Trim Galore! (v0.6.7) was used to filter the paired-end reads (Q > 30 and length > 50 bp) [23]. De novo assembly was performed using SKESA (v2.3.0; https://github.com/ncbi/SKESA (accessed on 3 April 2023)) and the assembled contigs were polished using Shovill (v1.1.0; https://github.com/tseemann/shovill (accessed on 3 April 2023)) [24]. Assembly metrics were calculated using QUAST (v5.0.2) [25]. The multilocus sequence typing (MLST) profiles were determined using the MLST tool (version 2.16.4; https://pubmlst.org/; https://github.com/tseemann/mlst (accessed on 3 April 2023)) [26]. The VF gene and ARG search was performed using ABRicate (version 1.0.1; https://github.com/tseemann/abricate (accessed on 3 April 2023)), against the Comprehensive Antibiotic Resistance Database (CARD), CARD-prevalence, Virulence Factor Database (VFDB), and ResFinder—Center for Genomic Epidemiology (CGE) database; with the gene alignment coverage cut-off of ≥95% and blastn sequence similarity of ≥95% [27,28,29,30,31,32]. A plasmid search was performed using PlasmidFinder (version 2.0; https://cge.food.dtu.dk/services/PlasmidFinder/ (accessed on 3 April 2023)). The assembled genome files were deposited in the National Center for Biotechnology Information GenBank under BioProject number PRJNA1006163.

### 3.4. Metagenomics

Total gDNA was extracted using the QIAamp Fast DNA stool mini kit (Qiagen, Hilden, Germany) and host depletion was performed using the NEBNext Microbiome DNA enrichment kit (New England Biolabs, Ipswich, MA, USA) with the inclusion of controls (the ZymoBIOMICS Gut Microbiome Standard was used as a positive control). Sequencing was performed on the NextSeq 550 (2 × 150 bp and 10 M reads) (Illumina, USA).

Initial sequence analysis was performed by the SCF, NICD. This included de-multiplexing (assigning reads to the respective sample using the barcodes that were assigned during the library preparation stage), quality checking using FastQC (https://www.bioinformatics.babraham.ac.uk/projects/fastqc/ (accessed on 15 April 2023)), and trimming and discarding of reads with a Q-score of less than 20 (TrimGalore, v0.6.2; https://github.com/FelixKrueger/TrimGalore (accessed on 15 April 2023)). Subsequent sequence analysis included the removal of host sequences (Suis scrofa v11.1 and *Homo sapiens* GRCh38.p13; https://www.ensembl.org/index.html (accessed on 15 April 2023)) using Bowtie2 v2.5.0 (https://github.com/BenLangmead/bowtie2 (accessed on 15 April 2023)). Taxonomic assignment was performed using the host-depleted metagenomic sequence reads, where the microbial diversity profile analyses were carried out with Kraken (v2.1; https://github.com/DerrickWood/kraken (accessed on 15 April 2023)) using the Standard-16 database [33]. This was followed by calculating the estimated related abundance of genera using Bracken (v2.5; https://ccb.jhu.edu/software/bracken/ (accessed on 15 April 2023)), and the proportion of related abundance was visualised using R (v4.2.1; https://www.r-project.org/ (accessed on 29 April 2023)). The assembled genome files were deposited in the National Center for Biotechnology Information GenBank under BioProject number PRJNA1137389.

### 3.5. Resistome Gene Abundance Estimates

With the host-depleted metagenomic sequence reads, standard de novo assembly of the metagenomic data was performed using MEGAHIT (v1.2.9; https://github.com/voutcn/megahit (accessed on 16 June 2023)) [34]. Using the assembled contigs, the ARGs, toxin genes, and VF genes were predicted using the PathoFact pipeline (v1.0; https://github.com/samnooij/PathoFact (accessed on 18 June 2023)) [35]. The predicted ARGs, as well as the toxin genes and VF genes with high confident prediction (1: secreted toxin or 1: secreted VF), were selected for relative abundance analysis, using ShortBRED (v0.9.4; https://github.com/biobakery/shortbred (accessed on 25 June 2023)) [36] against the CARD database (v2023-06) and VFDB (database update: 6 August 2023). The predicted resistome relative abundance was quantified by ShortBRED-Quantify calls USEARCH, where reads with 95% identity to the resistome were counted and normalized by reads per kilobase of reference sequence per million sample reads (RPKMs). The antimicrobial resistance (AMR) category “multidrug” was defined as bacterial strains that have become resistant to multiple classes of antibacterial drugs or other agents (https://card.mcmaster.ca/ontology/41472 (accessed on 25 June 2023)).

## 4. Discussion

The overuse and misuse of antibiotics in livestock production systems have led to residual contamination in food, resulting in antibiotic-resistant bacteria and ARGs [3,4]. In this study, we investigated and categorised four indicator bacteria (*Campylobacter* spp., *Enterococcus* spp., *E. coli*, and NTS spp.) that overlap between human and animals. We examined the diversity and abundance of bacterial communities, as well as ARGs and antibiotic residues in raw pork meat samples.

Out of the 10 raw meat samples collected, six did not contain any of the tested indicator organisms. A single *E. coli* isolate was obtained from sample PC9-B4, while one *E. faecalis* isolate was obtained from each of the samples PC3-S3, PC4-S4, and PC10-B5. The isolated *E. coli* and *E. faecalis* isolates showed minimal phenotypic resistance. The *E. coli* isolate showed resistance to tigecycline and trimethoprim/sulfamethoxazole, while all three *E. faecalis* isolates showed resistance to moxifloxacin only. WGS data for the *E. coli* (PC9-B4) and *E. faecalis* isolates (PC3-S3, PC4-S4, and PC10-B5) showed mainly the presence of tetracycline resistance genes.

International studies have shown that *E. faecalis* is the most dominant *Enterococcus* spp. isolated from pork samples (>80%) [37,38,39,40]. High tetracycline resistance and associated resistance genes found in this study are consistent with other reports [37,38,39,40,41]. In contrast, Aslam et al. (2012) found additional ARGs in enterococci isolated from retail meats in Canada, including genes for aminoglycosides (*aac*, *aph*A3, *aad*-A and -E, and *sa*t4), macrolides (*erm*-B and -A), streptogramin (*vat*E), bacitracin (*bcr*R), and lincosamide (*lin*B) [37]. Similarly, Hart et al. (2004) reported widespread tetracycline resistance in *E. coli* isolated from pigs in Australia [42]. The broad-spectrum antibiotic tetracycline, used to treat various infections and promote animal growth, has contributed to the high tetracycline resistance rates observed in this study [8,10,43,44,45].

The WGS data further revealed eight and nine different VF gene categories associated with the sequenced *E. coli* (PC9-B4) and *E. faecalis* (PC3-S3, PC4-S4, and PC10-B5) isolates, respectively. VF genes enable pathogenic bacteria to colonise host niches and establish infections, contributing both directly and indirectly to disease processes [35]. Aslam et al. (2012) reported that cytolysin (*cyl*-A, -B, -L and -M) and aggregation substances (*agg*) are associated with ARGs in retail meats, although the latter were not detected in our study [37,46]. Other VF genes, such as *ace*, *ebp*, and *efa*A have been reported in *E. faecalis* isolates from livestock and raw meat in Ghana and hospitalised patients in Iran [40,47].

The *E. coli* ST10 detected in this study is a known reservoir for ARG and mobile genetic elements, such as class 1 integrons and plasmids [45,48,49]. The IncB/O/K/Z and *Inc*FII (pCoo) plasmids detected in the *E. coli* isolates have also been reported in *E. coli* from hospitalised patients in South Africa and in carbapenem-resistant Enterobacterales from hospitalised patients in Thailand [50,51]. To our knowledge, *E. faecalis* ST30 has only been identified in hospitalised patients [52,53,54]. The repUS43, repUS11, and rep9a plasmids detected in the *E. faecalis* isolates have been reported in other *E. faecalis* isolates from livestock and raw meat in Ghana as well as hospitalised patients in the USA [40,55]. The repUS43 plasmid is integrated chromosomally near the tetracycline resistance gene (*tet*M), while the rep9a plasmid encodes the cytolysin VF gene [55]. Another study also reported that the *tet*-M and -L and *erm*B resistance genes were harboured on the repUS43 and rep9a plasmids [40]. Therefore, sequencing and plasmid monitoring are essential for future surveillance studies to track the spread and evolution of these genetic elements in humans, animals, and food products.

The antibiotic residue concentrations in the 10 tested raw meat samples were within acceptable limits for human consumption according to the published guidelines [12]. Ramatla et al. (2017) reported higher antibiotic residue concentrations in raw meat samples collected in North West, South Africa [10]. The discrepancy between their findings and ours could be due to differences in sampling frequency and sample types.

The metagenomic data revealed that most of the identified bacteria were associated with food/meat spoilage and environmental sources [16,17,18,19,20]. Their pathogenicity or ability to cause disease was not confirmed in this study. Resistome analysis revealed the presence of beta-lactam, tetracycline, and multidrug resistance genes in the 10 raw meat samples, consistent with previous local and internal studies [44,56,57].

Further research and analysis are needed to fully understand the potential implications of these findings in terms of environmental health and antibiotic resistance. This study has several limitations. Firstly, a high number of host reads (i.e., pig DNA) in the metagenomic data limited our ability to recover bacterial metagenome-assembled genomes (MAGs) from the raw meat samples. Consequently, this limited our ability to validate ARG, VF, and toxin gene predictions. Using the MAGs approach could provide more comprehensive information about the bacterial species in the meat samples. Secondly, increasing the sequencing depth or using alternative microbiome enrichment methods is recommended. Thirdly, the study’s limited sample size may affect the generalisability of the findings to other supermarkets and butcheries across South Africa.

## 5. Conclusions

In conclusion, this study highlights the importance of evidence-based investigation and laboratory testing for indicator bacteria and antibiotic susceptibility in food production systems. While none of the indicator bacteria were detected in significant colony-forming units in the raw meat samples, isolated *E. coli* and *E. faecalis* strains exhibited minimal phenotypic resistance. Notably, they exhibited resistance to tigecycline, trimethoprim/sulfamethoxazole, and moxifloxacin. WGS data highlighted prevalent tetracycline resistance genes, consistent with other studies on tetracycline use in pork production. The presence of various ARGs and VF genes in these supermarket and butchery meat samples highlight the need for effective antimicrobial practices in pork production. To prevent contamination by antibiotic-resistant bacteria, supermarkets and butcheries selling pork products should rigorously monitor microbial levels and adhere to sanitation standards. Although antibiotic residue concentrations remain within acceptable limits, further research is needed to understand the broader implications of these findings for environmental health and antibiotic resistance. Overall, this study emphasises the need for vigilance and comprehensive strategies to combat antibiotic resistance in food production systems.

## Figures and Tables

**Figure 1 antibiotics-13-00700-f001:**
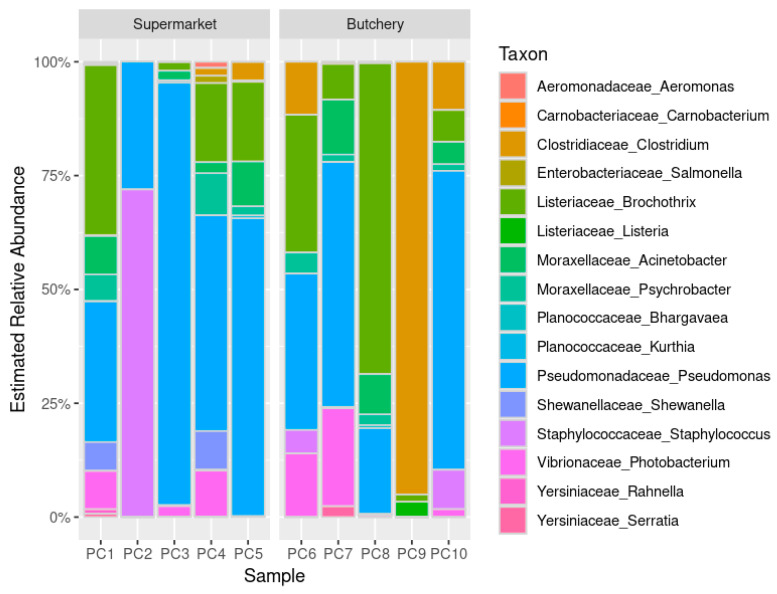
Estimated relative abundance of the bacterial taxonomic profiling (aggregated by family and genus) in 10 raw meat samples. PC = pork chop; PC1–5 = sampled from supermarkets; PC6–10 = sampled from butcheries.

**Figure 2 antibiotics-13-00700-f002:**
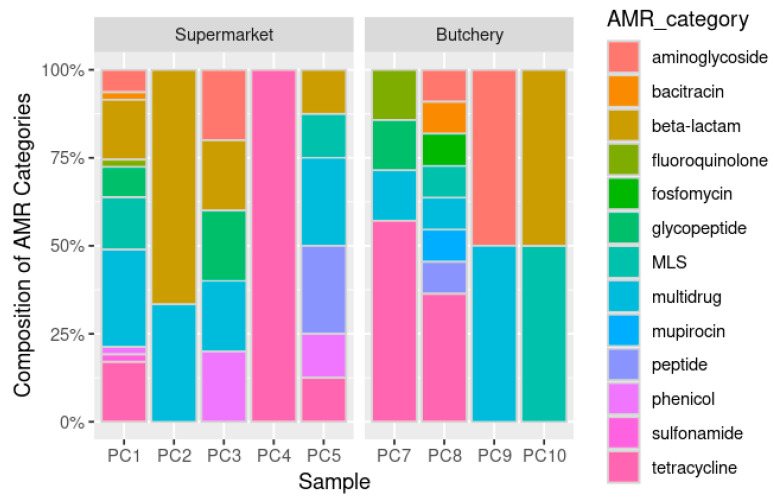
Composition of predicted AMR categories in 10 raw meat samples as predicated by PathoFact database. MLS = macrolides, lincosamides, and streptogramins; AMR category = default assignment by the PathoFact pipeline; PC6 = no AMR was detected.

**Figure 3 antibiotics-13-00700-f003:**
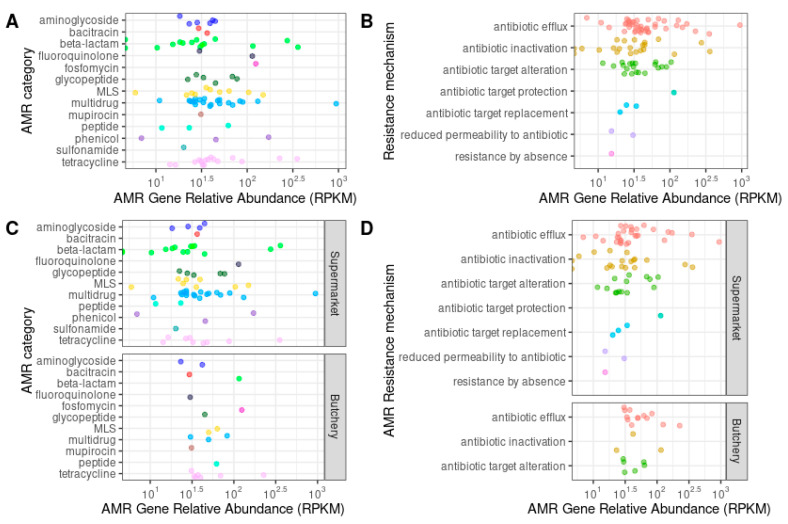
Relative abundance of individual ARGs of 10 raw meat samples (normalized as reads per kilobase of reference sequence per million mapped sample reads (RPKMs)), grouped by the AMR category (**A**,**C**), and by AMR resistance mechanisms (**B**,**D**), stratified by supermarket and butchery (**C**,**D**). Each circle indicating the relative abundance of an ARG gene in a single sample and genes within the same AMR category or AMR resistance mechanism are represented with the same colour.

**Figure 4 antibiotics-13-00700-f004:**
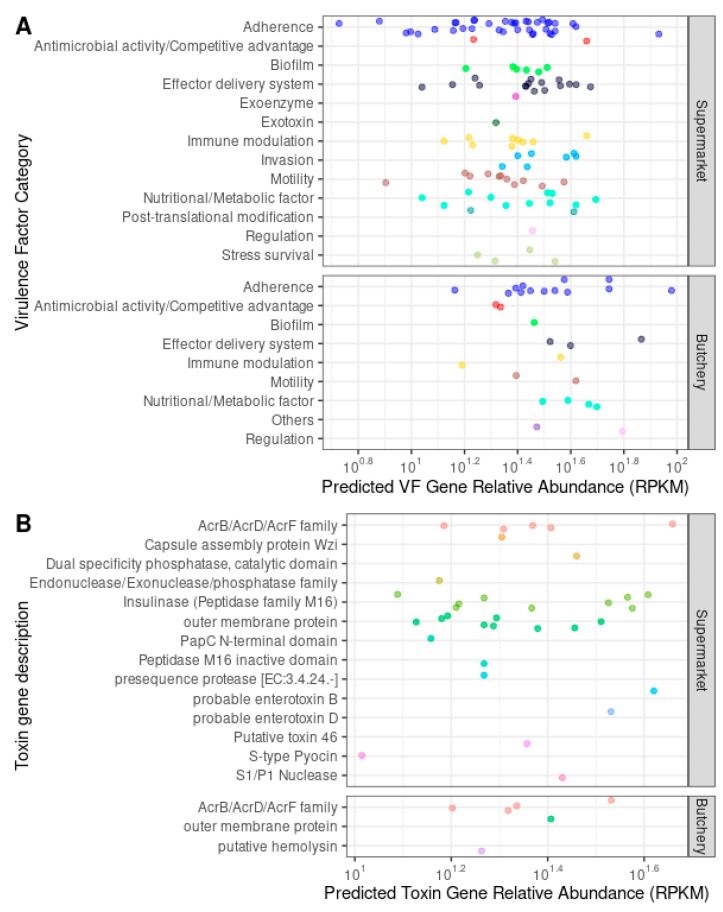
Relative abundance of the individual predicted virulence factors (VFs) and toxin genes in 10 raw meat samples (normalized as reads per kilobase of reference sequence per million mapped sample reads (RPKMs)), grouped by the VF category according to VFDB (**A**), and by toxin gene description according to the PathoFact database (**B**). Each circle indicates the relative abundance of a VF/toxin gene in a single sample. Genes within the same VF category or toxin gene description are represented with the same colour.

**Table 1 antibiotics-13-00700-t001:** Indicator organisms isolated from raw meat samples.

Sample ID	*E. coli* (CFUs)	*Salmonella* spp. (CFUs)	*Enterococci* spp. (CFUs)	*Campylobacter* spp. (CFUs)
PC1-S1	Absent	Absent	Absent	Absent
PC2-S2	Absent	Absent	Absent	Absent
PC3-S3	Absent	Absent	16 *	Absent
PC4-S4	Absent	Absent	1 *	Absent
PC5-S5	Absent	Absent	Absent	Absent
PC6-B1	Absent	Absent	Absent	Absent
PC7-B2	Absent	Absent	Absent	Absent
PC8-B3	Absent	Absent	Absent	Absent
PC9-B4	20 *	Absent	Absent	Absent
PC10-B5	Absent	Absent	3 *	Absent

PC = pork chop; S = supermarket; B = butchery; CFUs = colony-forming units; * = isolated bacteria from raw meat samples were subsequently processed (AST and WGS).

**Table 2 antibiotics-13-00700-t002:** Antibiotic susceptibility testing of one *Escherichia coli* isolate and three *Enterococcus faecalis* isolates isolated from four raw meat samples.

Antibiotic Class	Antibiotic	*E. coli* PC9-B4 (µg/mL)	*MIC Interpretation ^#^*	*E. faecalis* PC3-S3 (µg/mL)	*E. faecalis* PC4-S4 (µg/mL)	*E. faecalis* PC10-B5 (µg/mL)	*MIC Interpretation ^#^*
Aminoglycoside	Amikacin	≤8	S	32	32	32	NI *
	Gentamicin	≤2	S	4	4	4	NI *
	Gentamicin synergy	NT	-	≤500	≤500	≤500	NI
	Streptomycin synergy	NT	-	≤1000	≤1000	≤1000	NI
	Tobramycin	≤2	S	≤2/38	≤2/38	≤2/38	NI *
Beta-lactam (penicillins)	Ampicillin	≤8	S	4	4	4	S
	Ampicillin/sulbactam	≤8/4	S	NT	NT	NT	-
	Amoxicillin/clavulanic acid	≤8/4	S	≤4/2	≤4/2	≤4/2	S
	Oxacillin	NT	-	>2	>2	>2	NI
	Penicillin	NT	-	8	8	8	NI
	Piperacillin	≤8	S	NT	NT	NT	-
	Piperacillin/tazobactam	≤8	S	NT	NT	NT	-
Beta-lactam (cephalosporins)	Cefepime	≤1	S	NT	NT	NT	-
	Cefotaxime	≤1	S	NT	NT	NT	-
	Cefotaxime/clavulanic acid	≤0.5	NI	NT	NT	NT	-
	Cefoxitin	≤8	S	NT	NT	NT	-
	Cefuroxime	≤4	S	NT	NT	NT	-
	Ceftazidime	≤1	S	NT	NT	NT	-
	Ceftazidime/clavulanic acid	≤0.25	NI	NT	NT	NT	-
	Cephalothin	≤8	NI	NT	NT	NT	-
Beta-lactam (carbapenems)	Doripenem	≤1	S	NT	NT	NT	-
	Ertapenem	≤0.5	S	NT	NT	NT	-
	Imipenem	≤1	S	≤4	≤4	≤4	S
	Meropenem	≤1	S	NT	NT	NT	-
Beta-lactam (monobactams)	Aztreonam	≤1	S	NT	NT	NT	-
Amphenicol	Chloramphenicol	≤8	S	≤8	≤8	≤8	NI
Cyclic lipopeptide	Daptomycin	NT	-	≤1	≤1	≤1	NI
Fluoroquinolone	Ciprofloxacin	≤0.5	S	≤1	≤1	≤1	S
	Levofloxacin	≤1	S	≤1	≤1	≤1	S
	Moxifloxacin	NT	-	≤256	≤256	≤256	R
	Norfloxacin	≤0.5	S	≤4	≤4	≤4	NI
Fusidane	Fusidic acid	NT	-	≤2	≤2	≤2	NI
Lincosamides	Clindamycin	NT	-	>2	>2	>2	NI
	Pristinamycin	NT	-	2	2	2	NI
Macrolide	Erythromycin	NT	-	1	1	1	NI
Protein synthesis inhibitor	Mupirocin	>16	NI	NT	NT	NT	-
Nitrofuran	Nitrofurantoin	≤32	S	≤32	≤32	≤32	S
Phosphonic acid	Fosfomycin	≤32	S	≤32	≤32	≤32	NI
Polymyxin	Colistin	≤2	S	NT	NT	NT	-
Rifamycin	Rifampin	NT	-	≤0.5	≤0.5	≤0.5	NI
Tetracycline	Minocycline	>8	NI	≤1	≤1	≤1	NI
	Tetracycline	>8	NI	8	8	8	NI
	Tigecycline	≤1	R	NT	NT	NT	-
Glycopeptide and lipoglycopeptide	Teicoplanin	NT	-	≤1	≤1	≤1	S
	Vancomycin	NT	-	2	2	2	S
Oxazolidinone	Linezolid	NT	-	2	2	2	S
Sulfonamide	Trimethoprim/sulfamethoxazole	>4/76	R	NT	NT	NT	-

PC = pork chop; S = supermarket; B = butchery; NT = not tested (antibiotic not included in the antibiotic panel); NI = no EUCAST MIC interpretation; * = all *E. faecalis* are intrinsically resistant to aminoglycosides; ^#^ = [15].

**Table 3 antibiotics-13-00700-t003:** Antibiotic resistance genes, virulence factors, and plasmids detected in one *Escherichia coli* isolate and three *Enterococcus faecalis* isolates isolated from four raw meat samples.

Sample ID			PC9-B4 *	PC3-S3 *	PC4-S4 *	PC10-B5 *
Organism			*E. coli*	*E. faecalis*	*E. faecalis*	*E. faecalis*
CH type			11–54	-	-	-
O type			O69	-	-	-
H type			H32	-	-	-
MLST			10 ^	30 ^#^	30 ^#^	30 ^#^
ARGs	Aminoglycoside	*aad*A1	Y	-	-	-
	Fluoroquinolone	*gyr*A	Y	-	-	-
	Lincosamide	*isa*A	-	Y	Y	Y
	Sulphonamide	*sul*2	Y	-	-	-
	Tetracycline	*tet*B	Y	-	-	-
		*tet*M	-	Y	Y	Y
	Trimethoprim	*dfr*A1	Y	-	-	-
VF genes	Adhesin	*ace*	-	Y	Y	Y
		*efaAfs*	-	Y	Y	Y
	Colicin	*cba*	Y	-	-	-
		*cea*	Y	-	-	-
		*cia*	Y	-	-	-
		*cma*	Y	-	-	-
	Cytolysin toxin	*cyl*A	-	Y	Y	Y
		*cyl*L	-	Y	Y	Y
		*cyl*M	-	Y	Y	Y
	Endocarditis and biofilm-associated pili genes	*ebp*A	-	Y	Y	Y
		*ebp*B	-	Y	Y	Y
	*Enterococcus faecalis* leucine-rich protein A	*elr*A	-	Y	Y	Y
	Glutamate decarboxylase	*gad*	Y	-	-	-
		*gel*E	-	Y	Y	Y
	Heat stable toxin	*ast*A	Y	-	-	-
	Hyaluronidase	*hyl*A	-	Y	Y	Y
	Increased serum survival	*iss*	Y	-	-	-
	Outer membrane protease	*omp*T	Y	-	-	-
	Plasmid-encoded catalase peroxidase	*kat*P	Y	-	-	-
	Sex pheromone	*cad*	-	Y	Y	Y
		*cam*E	-	Y	Y	Y
		*cCF10*	-	Y	Y	Y
		*cOB1*	-	Y	Y	Y
	Tellurium ion resistance	*ter*C	Y	-	-	-
	Thiol peroxidase	*tpx*	-	Y	Y	Y
	Outer membrane protein complement resistance	*tra*T	Y	-	-	-
	Sortase	*Srt*A	-	Y	Y	Y
Plasmids		IncB/O/K/Z	Y	-	-	-
		IncFII(pCoo)	Y	-	-	-
		repUS43	-	Y	Y	Y
		repUS11	-	Y	Y	Y
		rep9a	-	Y	Y	Y

PC = pork chop; S = supermarket; B = butchery; ARG = antibiotic resistance genes; VF = virulence factor; MLST = multilocus sequence typing (https://github.com/tseemann/mlst (accessed on 3 April 2023)); Y = present/detected; - = absent/not detected; ^ = *adk* 10/*fum*C 11/*gyr*B 4/*icd* 8/*mdh* 8/*pur*A 8/*rec*A 2; ^#^ = *aro*E 10/*ghd* 7/*gki* 1/*gyd* 1/*pst*S 11/*xpt* 2/*yqi*L 1; * = the corresponding raw meat sample numbers were assigned to each isolate for traceability.

**Table 4 antibiotics-13-00700-t004:** Antibiotic residue testing in 10 raw meat samples using liquid chromatography–tandem mass spectrometry (LC-MS/MS).

Antibiotic Class	Antibiotic	Sample ID										Acceptable Maximum Residue Level (µg/kg)
		PC1-S1 * (µg/kg)	PC2-S2 * (µg/kg)	PC3-S3 * (µg/kg)	PC4-S4 * (µg/kg)	PC5-S5 * (µg/kg)	PC6-B1 * (µg/kg)	PC7-B2 * (µg/kg)	PC8-B3 * (µg/kg)	PC9-B4 * (µg/kg)	PC10-B5 * (µg/kg)	
Fluoroquinolones	Ciprofloxacin	<50	<50	<50	<50	<50	<50	<50	<50	<50	<50	100 ^#^
	Enrofloxacin	<50	<50	<50	<50	<50	<50	<50	<50	<50	<50	100 ^#^
	Norfloxacin	<50	<50	<50	<50	<50	<50	<50	<50	<50	<50	-
Lincosamides	Lincomycin	<50	<50	<50	<50	<50	<50	<50	<50	<50	<50	200 ^
Macrolides	Tylosin	<50	<50	<50	<50	<50	<50	<50	<50	<50	<50	100 ^
Sulfonamides	Sulfadiazine	<50	<50	<50	<50	<50	<50	<50	<50	<50	<50	-
	Sulfadimidine	<50	<50	<50	<50	<50	<50	<50	<50	<50	<50	100 ^
	Sulfamethoxazole	<50	<50	<50	<50	<50	<50	<50	<50	<50	<50	-
Tetracyclines	Chlortetracycline	<50	<50	<50	<50	71.5	<50	<50	<50	<50	<50	200 ^
	Doxycycline	<50	<50	<50	<50	<50	<50	<50	<50	<50	<50	-
	Oxytetracycline	<50	<50	<50	<50	<50	<50	<50	<50	<50	<50	200 ^
	Tetracycline	<50	<50	<50	<50	<50	<50	<50	<50	<50	<50	200 ^/600 ^#^
Pleuromutilin	Tiamulin	<50	<50	<50	<50	<50	<50	<50	<50	<50	<50	-
Diaminopyrimidines	Trimethoprim	<50	<50	<50	<50	<50	<50	<50	<50	<50	<50	-
Quindoxin	Olaquindox metabolite	<50	<50	<50	<50	<50	<50	<50	<50	<50	<50	-

PC = pork chop; S = supermarket; B = butchery; * = muscle; - = no acceptable maximum residue level listed; ^ = [12]; ^#^ = [10].

**Table 5 antibiotics-13-00700-t005:** Metagenomics read statistics of 10 raw meat samples.

Sample ID	Raw Paired-End Reads (*n* = NGS Reads)	Paired-End Reads after Host Removal (*n* = NGS Reads)	Paired-End Reads Mapped to Bacteria (*n* = NGS Reads)	Predicted ARG (*n* = Annotated ORF)	Predicted Secreted VF Genes (*n* = Annotated ORF)	Predicted Secreted Toxin Genes (*n* = Annotated ORF)
PC1-S1	6,559,325	761,775	249,495	47	86	24
PC2-S2	7,276,334	372,205	4242	6	2	5
PC3-S3	9,008,555	407,827	59,159	5	11	2
PC4-S4	8,346,930	353,759	25,353	1	0	0
PC5-S5	7,706,014	423,019	48,284	8	10	3
PC6-B1	6,948,972	260,638	2460	0	1	0
PC7-B2	6,318,244	388,517	47,285	7	12	3
PC8-B3	6,811,831	400,456	71,941	11	15	3
PC9-B4	8,083,308	336,636	2049	2	0	0
PC10-B5	6,727,542	295,466	2043	2	1	0

PC = pork chop; S = supermarket; B = butchery; NGS = next-generation sequencing; ARG = antibiotic resistance genes; ORF = open reading frame/s; VF = virulence factor.

## Data Availability

The assembled genome files were deposited in the National Center for Biotechnology Information GenBank and are available under the following BioProject numbers: PRJNA1006163 and PRJNA1137389. The Appendix A contain additional data.

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
