# Peer review of "Bacterial and Genetic Features of Raw Retail Pork Meat: Integrative Analysis of Antibiotic Susceptibility, Whole-Genome Sequencing, and Metagenomics"

_antibiotics, 2024, doi:10.3390/antibiotics13080700_

Round 1

Reviewer 1 Report

Comments and Suggestions for Authors

Lowe et al, Antibiotics July 2024.

Review of "Bacterial and Genetic Features of Raw Retail Pork Meat:..."

The authors address the 'microbiological and genetic characteristics of ' raw pork, bought at supermarkets and butcheries in various stores in South Africa. They screened the pork samples for indicator bacteria, residual antibiotics and also carried out WGS/metagenomic sequencing on the pork samples. These studies showed the presence of tetracycline resistance genes in the indicator bacteria, little evidence of antibiotics and, from the metagenomics approaches, the presence of bacteria normally associated with food spoilage. Some of these bacteria carried antibiotic resistance genes and virulence factors.

This is a high quality study, using state-of-the art approaches and analyses. While the study is limited by the small number of samples, this is a valuable addition to the other published surveys of bacteria and antibiotics present in commercially-available foodstuffs. The paper is well-written and I have no further comments for its improvement.

Author Response

1. Summary

Thank you very much for taking the time to review this manuscript.

2. Point-by-point response to Comments and Suggestions for Authors

No comments were received from Reviewer 1.

3. Additional clarifications

Please note that the section managing editor requested changes to the Results and Materials and Methods sections. The Results now appear before the Materials and Methods section.

Reviewer 2 Report

Comments and Suggestions for Authors

This study addresses the global antibiotic resistance crisis by examining the microbiological and genetic characteristics of raw retail pork meat from 10 stores in Gauteng, South Africa. The research included antibiotic susceptibility testing (AST), whole genome sequencing (WGS), and metagenomic sequencing of isolated bacteria. The isolated E. coli and Enterococci showed minimal phenotypic resistance, but WGS identified tetracycline resistance genes in both the bacteria and meat samples. Metagenomic analysis revealed bacteria mainly from spoilage and environmental origins, with resistance genes predominantly for beta-lactam, tetracycline, and multidrug resistance. The manuscript is well-written and adequately demonstrated. Nevertheless, there are areas that could be enhanced to uphold the quality of the research article.

Introduction

1.     Authors should update the literature review with recent studies and key references to strengthen the manuscript.

Methods

2.     In the methodology section, the authors should specify the criteria used for random store selection.

Results

3.     Authors should include statistical analysis (confidence intervals, p-values) to support findings.

4.     Table 4: Add units for acceptable maximum residue levels for clarity.

Author Response

Response to Reviewer 2 Comments

1. Summary

2. Point-by-point response to Comments and Suggestions for Authors

Comments 1: Authors should update the literature review with recent studies and key references to strengthen the manuscript.

Response 1: The introduction has been updated.

Please see lines 35-72 (without track changes).

Comments 2: In the methodology section, the authors should specify the criteria used for random store selection.

Response 2: The methodology section was updated - “Raw meat samples (i.e. pork chops) were purchased on the 4th of January 2022, from 10 pre-selected supermarkets and butcheries in Johannesburg and Pretoria, Gauteng. Convenience sampling was used based on the location of the laboratory, the Centre for Healthcare-associated infections, Antimicrobial Resistance and Mycoses (CHARM)”.

Please see lines 208-211 (document without track changes).

Comments 3: Authors should include statistical analysis (confidence intervals, p-values) to support

Response 3: One limitation of our study is the small sample size (n=10), which limits the statistical analysis that could be applied. We conducted a Kruskal-Wallis test using the ‘rstatix’ package (v0.7.2) to assess whether differences between the supermarkets and butcheries were significant. The results showed statistically significant differences in the relative abundance of antibiotic resistance genes (AMR) and virulence factor genes (VF) across different meat sources (p=0.03 and p=0.00061, respectively). However, there was no statistically significant difference in the relative abundance of toxin genes between the different meat sources.

Please see Supplementary Figure 1.

Comments 4: Table 4: Add units for acceptable maximum residue levels for clarity.

Response 4: Table 4 has been updated - the units have been added as requested.

Please see Table 4.

3. Additional clarifications

Please note that the section managing editor requested changes to the Results and Materials and Methods sections. The Results now appear before the Materials and Methods section.

Reviewer 3 Report

Comments and Suggestions for Authors

1. The methodology section should be enhanced by including the methods used to identify plasmid sequences from genome sequences.

2. The authors should provide the submission IDs of the 10 metagenome sequences.

3. Minor comments and suggestions are provided in the attached file (highlighted in the manuscript text).

Author Response

Response to Reviewer 3 Comments

1. Summary

2. Point-by-point response to Comments and Suggestions for Authors

Comments 1: Mention the number of isolates used for gDNA extraction? Based on the PRJNA1006163, it is four.

Response 1: “all” was changed to “four”. “The genomic DNA (gDNA) of four isolated organisms from the raw meat samples were extracted with the QIAamp mini kit (Qiagen, Germany) with the inclusion of lysozyme (10 mg/mL; Sigma-Aldrich, USA) to ensure sufficient lysis”.

Please see lines 241-243 (document without track changes).

Comments 2: Which method was used for the gDNA fragmentation? Please mention it.

Response 2: The methods section was updated. “The tagging and fragmentation of the gDNA were done using the Nextera DNA Flex Library preparation kit”.

Please see lines 245-246 (document without track changes).

Comments 3: Italics

Response 3: Homo sapiens was written in italics.

Please see line 278 (document without track changes).

Comments 4: How did authors identify the plasmids and their completeness? Mention tool(s) used to detect the plasmids from WGS seqs and their locations in the genomes. Based on the https://github.com/stanikae/jekesa pipeline, plasmid detection methodology is under construction. Did authors use Plasmid Finder or PLSDB database? Authors are suggested to mention the methodology/ tool used in the methodology section.

Response 4: The methods section was updated. “A plasmid search was performed using PlasmidFinder (version 2.0; https://cge.food.dtu.dk/services/PlasmidFinder/)”.

Please see lines 261-262 (document without track changes).

Comments 5: Is it 71.5 micro-gram/Kg? The reader from other continents might confuse.

Response 5: The comma was replaced with a point (71,5 à 71.5).

Please see line 120 and Table 4.

Comments 6: Italics

Response 6: sat was written in italics.

Please see line 321.

Comments 7: Also, to avoid AMR microbe contamination, butcher shops and supermarkets that sell pork products must maintain high standards of sanitation and cleanliness. Such point can be added after the highlighted sentence.

Response 7: This was added to the conclusion section. “To prevent contamination by antibiotic resistant bacteria, supermarkets and butcheries selling pork products should rigorously monitor microbial levels and adhere to sanitation standards”.

Please see lines 379-381.

Comments 8: It is for 4 WGS sequencing. The authors should provided the submission ID of 10 metagenome sequences.

Response 8: The submission reference for the metagenome sequences was added. This was missed by the authors and not included in the previous version of the manuscript. “PRJNA1006163 and PRJNA1137389”.

Please see lines 285-286 and line 415.

3. Additional clarifications

Please note that the section managing editor requested changes to the Results and Materials and Methods sections. The Results now appear before the Materials and Methods section.

Reviewer 4 Report

Comments and Suggestions for Authors

The authors utilized both innovative technologies, such as metagenomic analysis, and conventional methods to characterize specific bacteria in meat products. As the authors mention, there are several limitations to the study, including the small sample size and the high number of host reads despite performing host-depletion on the samples. Nevertheless, it is a well-structured manuscript and the presentation of findings is comprehensive.

I recommend reviewing the discussion section, as several areas need improvement to enhance clarity and coherence. Additionally, please include a comment about ST30 E. faecalis in comparison with existing literature.

How many paired-end reads does your metagenomic analysis produce? Can you explain why, despite having several hundred thousand reads, only a small proportion belong to bacteria?

In section 3.4.3, to which multidrug-resistant genes are you referring? Please add a comment for clarification.

Comments on the Quality of English Language

Minor editing in the Discussion is recommended

Author Response

Response to Reviewer 4 Comments

1. Summary

2. Point-by-point response to Comments and Suggestions for Authors

Comments 1: I recommend reviewing the discussion section, as several areas need improvement to enhance clarity and coherence.

Response 1: The discussion section was updated.

Please see lines 302-368 (document without track changes).

Comments 2: Additionally, please include a comment about ST30 E. faecalis in comparison with existing literature

Response 2: The requested information was added and discussed as requested.

Please see lines 33-348 (document without track changes).

Comments 3: How many paired-end reads does your metagenomic analysis produce? Can you explain why, despite having several hundred thousand reads, only a small proportion belong to bacteria?

Response 3: Read statistics are shown in Table 5 (raw paired-end reads, paired-end reads after host removal, and paired-end reads mapped to bacteria). The total number of raw paired-end reads produced falls within the range of 6 559325 to 9 008555. The high levels of host reads and low levels of bacterial reads could be attributed to the following factors: 1) Failed initial host depletion steps - The initial steps to remove host-related reads were not successful; 2) Sus scrofa reference genome mismatch - The Sus scrofa reference genome used for host removal might not closely match the specific South Africa Sus scrofa commercial species. Reads similar to the reference genome were removed but host-specific reads unique to the South Africa Sus scrofa commercial species could still remain and 3) The sequencing depth was not sufficient (100x-200x).

Comments 4: In section 3.4.3, to which multidrug-resistant genes are you referring? Please add a comment for clarification.

Response 4: Section 3.4.3 is now 2.4.3 (see additional clarifications sections. The term MDR genes was defined in section 3.5. “The antimicrobial resistance (AMR) category “multidrug” was defined as bacterial strains that have become resistant to multiple classes of antibacterial drugs or other agents (https://card.mcmaster.ca/ontology/41472)”.

Please see lines 298-300:

3. Additional clarifications

Please note that the section managing editor requested changes to the Results and Materials and Methods sections. The Results now appear before the Materials and Methods section.